# Recent Advancements in Electrochemical Biosensors for Monitoring the Water Quality

**DOI:** 10.3390/bios12070551

**Published:** 2022-07-21

**Authors:** Yun Hui, Zhaoling Huang, Md Eshrat E. Alahi, Anindya Nag, Shilun Feng, Subhas Chandra Mukhopadhyay

**Affiliations:** 1Shenzhen Institute of Advanced Technology, Chinese Academy of Sciences, Shenzhen 518055, China; yun.hui@siat.ac.cn; 2School of Mechanical and Electrical Engineering, Guilin University of Electronic Technology, Guilin 541004, China; zhaoling_huang@guet.edu.cn; 3Faculty of Electrical and Computer Engineering, Technische Universität Dresden, 01062 Dresden, Germany; anindya.nag@tu-dresden.de; 4Centre for Tactile Internet with Human-in-the-Loop (CeTI), Technische Universität Dresden, 01069 Dresden, Germany; 5State Key Laboratory of Transducer Technology, Shanghai Institute of Microsystem and Information Technology, Chinese Academy of Sciences, Shanghai 200050, China; 6The School of Science and Engineering, Macquarie University, Sydney 2109, Australia; subhas.mukhopadhyay@mq.edu.au

**Keywords:** biosensors, electrochemical detection, water quality monitoring, bio-recognition element, in-situ monitoring, surface modification

## Abstract

The release of chemicals and microorganisms from various sources, such as industry, agriculture, animal farming, wastewater treatment plants, and flooding, into water systems have caused water pollution in several parts of our world, endangering aquatic ecosystems and individual health. World Health Organization (WHO) has introduced strict standards for the maximum concentration limits for nutrients and chemicals in drinking water, surface water, and groundwater. It is crucial to have rapid, sensitive, and reliable analytical detection systems to monitor the pollution level regularly and meet the standard limit. Electrochemical biosensors are advantageous analytical devices or tools that convert a bio-signal by biorecognition elements into a significant electrical response. Thanks to the micro/nano fabrication techniques, electrochemical biosensors for sensitive, continuous, and real-time detection have attracted increasing attention among researchers and users worldwide. These devices take advantage of easy operation, portability, and rapid response. They can also be miniaturized, have a long-life span and a quick response time, and possess high sensitivity and selectivity and can be considered as portable biosensing assays. They are of special importance due to their great advantages such as affordability, simplicity, portability, and ability to detect at on-site. This review paper is concerned with the basic concepts of electrochemical biosensors and their applications in various water quality monitoring, such as inorganic chemicals, nutrients, microorganisms’ pollution, and organic pollutants, especially for developing real-time/online detection systems. The basic concepts of electrochemical biosensors, different surface modification techniques, bio-recognition elements (BRE), detection methods, and specific real-time water quality monitoring applications are reviewed thoroughly in this article.

## 1. Introduction

Water is an essential part of all the living beings on earth, but in recent times, anthropogenic activities have increased immensely, which are the major causes of water pollution, disturbing the marine biodiversity and leading to a tremendous water shortage [1,2,3]. Even though the chemicals and water nutrients are crucial to our day-to-day lives, the excessive amount threatens humans, aquatic life, and animals. The pollution of water and habitat degradation are the causes of the escalating water shortage and the reasons for the deterioration in marine biodiversity. Although freshwater accessibility has deteriorated over the past decades, water demand has risen, particularly in warm areas with minimal rainfall. Recently, 71% of the world’s inhabitants, equal to 4.3 billion, were dealing with water shortages for several months [4]. Although water demand sharply increased, massive water pollution increased water scarcity and declining water quality in the past decades.

The characteristics of water pollution are comprised of their physical presence, chemical parameters, and richness of microorganisms. The concentration and composition of ingredients in water differ extensively. They can be categorized into four distinct classifications, such as (i) inorganic chemicals, (ii) nutrients, (iii) microorganisms’ pollution, and (iv) organic pollutants. They can bring about harmful ecological consequences, for example, the interference of internal secretion and hormone systems, stimulation of genotoxicity and cytotoxicity, and hazardous effects [5]. The strength of ingredients in water is essential for selecting, designing, and operational treatment processes and recycling waste. The variable quantity of contaminants in effluent over time also increases the attention to emerging technologies for monitoring the water and applying reasonably priced and real-time approaches [6]. This review is mainly focused on monitoring heavy metals, nutrients, organic pollutants, biochemical oxygen demand, and microorganisms. Heavy metals in soil and water are considered environmental contaminants with elevated toxicity, easy accretion, and complicated degradation [7]. Nutrients bring about water eutrophication. Organic pollutants, particularly persistent organic pollutants (POPs), have harshly harmful impacts on human health and the environment with their complex degradation and potential bioaccumulation [8]. The biochemical oxygen demand (BOD) is the essential supervisory index to measure organic water contamination and demonstrate water quality [9,10]. Water quality monitoring is critical and closely related to our life and production.

Conventional analytical techniques or laboratory-based procedures, such as gas chromatography (GC), high-performance liquid chromatography (HPLC), atomic absorption spectroscopy (AAS), atomic fluorescence spectrometry (AFS), and inductively coupled plasma mass spectrometry (ICP-MS), are sensitive, precise, and consistent. They are regularly used to measure water parameters with the help of trained operators. However, they are involved with bulky and costly instrumentation, take much time for sample preparation, and are unsuitable for in situ measurements, especially requiring trained operators’ help and transporting the water samples to laboratories for assessment [11,12,13]. Additionally, they cannot asses the accumulative toxicity or nutrient value of multiple chemicals or pollutants in a sample, which is a crucial objective of water quality monitoring applications [14]. Many property indicators are regularly used to determine the different qualities of water for settling or recycling. Many of them are laboratory-based techniques, which require complex pretreatment, and consequently, the methods are sluggish and expensive [1,15]. These characteristics encourage developing new technologies that are more low-cost, portable, sensitive, and efficient in the on-site real-time detection of multi-contaminants containing a wide variety of materials [16,17]. The significant challenges of developing a portable biosensing device are inadequate sensitivity and poor selectivity during the on-site detection. The significant level of noises can come on chemical components level from the sampling field and ambient environments can be variable due to the harsh environments or diurnal variations. These are the major obstacles where the researchers are putting lot of attentions on how to avoid these for generating a reliable and portable biosensing output signal. The portable biosensing method is successfully utilized for other applications, such as pesticide residues in fruits and vegetables [18], POC Detection for biomedical application [19], chemical and biological pollutants in water [20].

In recent years, the advancement of electrochemical biosensors for detecting environmental pollutants has received considerable attention [21,22,23,24,25]. Biosensors have many advantages over the conventional lab-based method, including low costs, portability, fast response time, less usage of reagents, and the capability to continuous monitor the complex wastewater [26,27,28]. Such sensors significantly benefit from sensing the minimum level in polluted water, such as wastewater. Biosensors are also compact and miniaturized devices that facilitate the advancement of portable sensing systems to monitor on-site effluents [29]. Bearing in mind the wide range of bio-recognition elements (including enzymatic, immunochemical, non-enzymatic receptor, whole-cell and DNA elements, and molecularly imprinted polymer (MIP)), the various types of biosensors can be classified as (i) electrochemical [30], (iii) piezoelectric [31], (ii) optical [32], and (iv) thermal biosensors [33] based on their working principles and transducing mechanisms [34], but the current review paper will cover the topics which are related to electrochemical biosensing. An electrochemical biosensor is based on the interactions between the immobilized bio-recognition element on its surface with binding molecules (the analyte of interest) and generating the changes in electrochemical properties, further translating into a meaningful electrical signal. The electrochemical methods offer rapid detection, fabrication, excellent sensitivity, and low cost.

Moreover, by operating at a wide range of potential, it is possible to simultaneously determine multiple analytes with different electrochemical potentials. Electrochemical biosensors’ efficiency in monitoring water pollutants’ presence relied on bio-recognition elements, transducers, and immobilization techniques, which offer us the classification criterion. In comparison with optical methods, electrochemical transduction has advantages for analyzing turbid samples because it is non-sensitive to light. For optical sensing, they are likely to be interference from environmental effects, costly, and susceptible to physical damage.

This review provides an overview of recent progress in developing electrochemical biosensors for water quality detection, focusing on the last decade. Some older publications are cited to support and build up the critical concepts of electrochemical biosensors. We expect this critical review will help those working in ecological toxicant analysis in water, some scientists who might be unaware of electroanalytical chemistry and biosensors.

## 2. Electrochemical Biosensors

Electrochemistry is essential for achieving the biosensing process in various biomarker analyses. Thus, electrochemical biosensing has attracted widespread attention in various applications due to its considerable advantages. Electrochemical biosensors react with the analyte of interest or molecules to produce an electrical signal proportionate to the analyte concentration. A conventional electrochemical biosensor comprises a reference electrode and a sensing electrode (working electrode) separated by an electrolyte. In most applications, the electrochemical biosensors consist of a three-electrode system with the reference electrode connected to a potentiostat, and the circuit can be completed by adding a counter electrode for flowing the current. These sensing devices are inexpensive, low-cost electrochemical cells that can be produced, portable, and easy to use, and can be operated with reduced power consumption. It requires electronic components for detecting the target analytes, unlike optical sensors. The following sections describe a range of elements and techniques of the electrochemical biosensor for biosensing applications.

## 3. Surface Modification Technique

Surface chemistry plays a considerable role in electrochemical biosensors to link the biological recognition element (BRE) on top of the sensing surface and prevent the substrate electrode from nonspecific interactions. In addition, the functionalization of the surface is conducive to noise control and sensitivity enhancement. BRE used in electrochemical sensors mainly consists of enzymes, antibodies, DNA/RNA, aptamers, and whole cells [22], which define a biosensor’s sensitivity and selectivity. Immobilization techniques, such as adsorption, encapsulation in polymers or gel, chemical crosslinking, self-assembled monolayer, covalent linking, affinity, and electrodeposition, have been widely investigated for detecting various analysts in complex water samples. The surface modification of BRE on the electrodes usually involves one or more strategies.

### 3.1. Adsorption

Adsorption is a straightforward method of modifying the surface of the electrode to a specific recognition element in an entirely arbitrary way. Every biological recognition element needs to achieve the best conditions. Most proteins usually achieve the best surface coverage on uncharged surfaces under the neutral pH and functional ionic strength, using a specific 5−20 μg/mL concentration [35]. Yang et al. [36] have developed an impedimetric-based immunosensor by the adsorption method of anti-*E. coli* antibodies against the integrated microelectrode arrays for detecting the *E. coli* O157:H7. Surface modification of the surface receptor proteins G and A can be produced by many bacterial strains that can promote receptor binding. Each protein, such as A and G, can certainly be capable of binding 4 and 2 molecules of IgG. An analogous method utilizes or takes advantage of the strong binding of the glycoprotein avidin for biotin-functionalizing the receptors due to the intense alienation of the avidin/biotin complex. The detection level of *E. coli* was 1.3 × 10^−15^ M, which was as low as 10−100 CFU mL^−1^ in concentration. It may be identified on the avidin-modified developed electrodes using biotinylated anti-*E. coli* as the targeted recognition ligand.

### 3.2. Self-Assembled Monolayers (SAMs)

Self-Assembled Monolayers (SAMs) are chemisorbed and ordered with various layers formed by the natural arrangement of thiolated molecules on the location of metallic interfaces. The most extensively used methods consist of SAMs with n-alkanethiols on noble metals [37], SAMs with carboxylate on the oxide surfaces [38], and SAMs with silane on the glass/silicon surfaces [39]. Xia et al., immersed the sidewall of the silica core into AuNSM colloid, forming a self-assembled AuNSM monolayer for sensitive wavelength-modulated localized surface plasmon resonance (LSPR) for detecting the mercury (II) [40]. The label-free sensor obtained a very low LOD of 0.7 nM owing to the near field coupling improvement by the proximity distance of two types of gold nanoparticles-DNA conjugates.

### 3.3. Covalent Attachment

Covalent attachment is another approach for the covalent coupling to the ligand recognitions to electrochemical biosensor’s interfaces, and improvements from the arrays of protein help form the most favorable conditions. A commonly used crosslinking molecule is carboxylic acid (C(=O) OH) groups on the electrode’s surface as the biorecognition element with amine functional groups for exploiting the amide bond formation using the techniques of EDC/NHS chemistry. Likewise, this coupling approach has been effectively applied in various three-dimensional supports, such as agarose, aldehyde−agarose, and carboxymethylated dextran-based modified electrodes [41]. Carbon-based materials that reduce graphene oxide and carbon nanotubes can be adjusted with carboxylic acid through π−π stacking interaction. Furthermore, some researchers have lately proposed integrating covalent functional groups using diazonium chemistry [42].

### 3.4. Electrodeposition

The electrochemical deposition was crucial in preparing nanomaterials reliably and cost-effectively with mild physicochemical conditions. Furthermore, noble metals, mixed metal oxides, carbon materials, or conducting polymers can be deposited on the electrode with high deposition speed, straightforward scale-up techniques and commercial feasibility with standard maintenance. This method helps form the hybrid films with the controlled thickness and morphology, modifying the process parameters, controlling the bath conditions (solvent, pH, temperature), and effectively regulating the electrolyte formula [43]. For example, new properties immediately stand out when poly(3,4-ethylenedioxythiophene) associates with one or more components deposited as films [44]. Table 1 shows the various characteristics of the surface modification techniques for the BRE in electrochemical biosensors.

## 4. Biorecognition Elements

Biorecognition elements (Figure 1) are the most critical part of the electrochemical biosensor, shouldering specific pollutants’ specific recognition in a complex matrix. The effective immobilization of biorecognition elements, including enzymes, antibodies, DNA/RNA, aptamers, and whole cells, facilitates their binding to a noticeably broad range of the target species or analyte of interest. As is shown in Table 2, different kinds of recognition elements were summarized, as well as the analytes, electrodes, type of transducers, and the response of various electrochemical biosensors. Apart from conserving the functionality of the bio-recognition elements, e.g., specific enzymatic activities, it is critical to ensure the biomaterials’ accessibility to target analytes. The vicinity between the biomaterials and the solid or metallic electrode surface is also preferred for achieving a fast and effective electron transfer. Several techniques have been suggested, including physical (e.g., electrostatic adsorption), chemical (e.g., self-assembled monolayers, covalent bonding, avidin-biotin binding, hybridization), and electrochemical (e.g., electrochemical adsorption) methods, but the optimal configuration for the biorecognition elements depends on the biomaterial and the modified electrode materials and interface. Besides the immobilization strategies, nonspecific adsorptions that mostly lead to high baseline signals and delayed responses should also be considered. Generally, various highly hydrophilic composites, for instance, poly (ethylene glycol) (PEG) and bovine serum albumin (BSA) and can be considered as additional elements for eliminating the nonspecific binding sites at the electrode and solution interface.

### 4.1. Enzyme-Based Bio-Recognition Materials

Enzym electrodes have been widely studied for the superior catalytic activity of inclusiveness of the enzymes and are commercially accessible. Regarding the electrochemical detection of water pollutants with enzymes, in some cases, they are converted to indirect detection of corresponding substrates by inhibiting enzyme activity. The biotransformation of various compounds can catalyze these enzymes through oxidation–reduction reactions (REDOX). The amperometric biosensors’ electrical responses of Oxidase/Peroxidase have electrical responses to a specific substrate that can be measured either by two different methods, such as direct or indirect methods [58]. Enzymes can be immobilized by physical adhesion or entrapped by the process of electrochemical techniques. According to the Michaelis–Menten equation, the enzymatic sensor’s detection limit depends on the enzyme’s activity. Thus, accommodating proper conditions (desirable temperature and buffer pH, etc.) for the development of the biosensor throughout the experiment is significant.

Ayenimo et al., rapidly developed a reliable, sensitive amperometric glucose biosensor to rapidly determine Hg^2+^, Cu^2+^, Pb^2+^, and Cd^2+^. A conductive ultrathin polypyrrole (PPy) film where the thickness was 55 nm thick was utilized to entrap glucose oxidase (GOx) with a fast response time. Upon exposure to trace metals, a more robust inhibition of GOx activity led to reduced glucose amperometric response [59]. Messaoud et al., have used fixed potential amperometry to determine bisphenol A (BPA), based on xanthine oxidase (XOD) enzymatic inhibition hypoxanthine as enzyme substrate, as is shown in Figure 1a,b. The mechanism of enzyme inhibition was estimated from the Cornish–Bowden and Dixon plots that were reversible with the competitiveness. An extremely low detection limit of 1.0 nM was achieved with excellent repeatability and reproducibility. The biosensor water samples’ selectivity, stability, and practical tests were also investigated [60].

### 4.2. Antibody

Immunoassay is based on detecting antigen–antibody conjugates or excessive other reagents (e.g., enzyme-labelled second antibody). Furthermore, it can be divided into competitive mode and non-competitive mode based on whether the analyte competes for a restricted number of binding antibody sites with the labelled analyte (e.g., indirect competitive immunoassay) or not (e.g., sandwich format) [61]. Immunoassays can also be classified as homogeneous and heterogeneous assays. Antibodies and antigens move freely from a complex immune situation to the solution phase in a homogenous format. However, it can be seen differently in a heterogeneous structure where the antibodies (or sometimes antigens) can be immobilized on a solid support to form the complex. Both types have been widely investigated, but homogeneous assays benefit from the possibility of multiplexing the complex format and separations are fast.

In contrast, the heterogeneous structure takes advantage of the elevated ratio of surface area to volume, which provides an additional higher sensitivity. Electrochemical immune sensors exhibit high sensitivity and selectivity compared to redox detection, which is extremely important in detecting various pesticides to decrease their mutual interference. The detection principle is mainly based on the current or impedance changes induced by antibody–antigen interaction, including chronoamperometry (CA) and electrochemical impedance spectroscopy (EIS).

A non-competitive immunoassay combined the with magneto-electrochemical immune sensors. It was developed to detect herbicide atrazine, one of the most used pesticides globally [62]. It is based on the recombinant M13 phage particles that bear a molecule named peptide. It is recognized explicitly as the immune complex of atrazine with an anti-atrazine monoclonal antibody. However, it is worth mentioning that each phage bore thousands of HRP Molecules, indicating the increased activity of pyrocatechol oxidation in the presence of hydrogen peroxide (H_2_O_2_). The phage anti-immunocomplex electrochemical immunosensor (PhAIEI) had dominant features, which provided a 200-fold improvement in sensitivity and a 10-fold wide linear working range compared with previous work with the same monoclonal antibody and anti-immunocomplex peptide. By chronoamperometry (CA), the fabricated PhAIEI was successfully applied in untreated river samples with excellent recoveries.

The leucomalachite green and malachite green in the water from a fish farm were detected by a BSA-decorated gold nanocluster (BSA-AuNC) with antibody composite film using the electrochemical impedance spectroscopy (EIS) method. The film was modified by a glassy carbon electrode (GCE). The modification was carried out due to the potential hazards to the human immune system and the human reproductive system [63]. Moreover, the BSA-AuNCs interface’s stability was improved via a diazotization method, and the antibody against leucomalachite green was chemically connected with the interface under the optimum conditions. After two weeks, the EIS immunosensor showed acceptable repeatability and stability with a negligible impedance reduction. A low LOD of 0.03 ng/mL was also obtained and compared with the ELISA method.

Azri et al., have developed an ultrasensitive electrochemical immunosensor for the detection of aflatoxin B1 (AFB1) based on an indirect competitive enzyme-linked immunosorbent assay (ELISA) to study the antigen–antibody interaction and optimize the optimum parameters of the assay [55]. The immunosensor demonstrated an excellent duplicability (RSD of 9%), and the response was logarithmic, where the detection range of 50–10,000 pM of IMD under the optimal conditions. The sensor was developed with the combination of BSA-labelled antigen and enzymatic tags. Compared with standard analytical methods, the developed sensor demonstrated a more comprehensive lower detection limit and a comprehensive range of responses which satisfied the detection requirements considering the European Union legislation. Saravanan et al., proposed a simple, disposable, and low-cost, paper-based immunosensor to detect bacteria in water [64]. The screen-printed fabrication technique was used for printing a conductive carbon electrode onto a commercial hydrophobic paper. Carboxyl groups were utilized for functionalization with the lectin Concanavalin A, which was covalently immobilized as the selective coating for biorecognition element for interacting with mono- and oligosaccharides. A linear calibration curve was developed for bacterial concentrations ranging from 10^3^~10^6^ CFU mL^−1^, with the projected lower detection limit of 1.9 × 10^3^ CFU mL^−1^.

Immunomagnetic assays with the introduction of magnetic beads (MBs) are particularly effective for enhancing the analytical performance. A huge surface area allows them to be utilized in immobilization of biomolecules, such as enzymes, DNA, and antibodies. Chemical and physical stability, low toxicity, and high biocompatibility make them suitable for the immobilization of biomolecules. Efficient dispersing ability enables them to shorten the reaction time between dissolved species and biomolecules [54]. The electrochemical glyphosate (N-(phosphonomethyl)glycine) biosensor has been developed with a disposable screen-printed electrochemical cell and applied to the analysis of spiked beer samples based on the competitive assay, as is shown in Figure 1c [54]. With tetramethylbenzidine (TMB) as the enzymatic substrate, the affinity reaction’s scope has been achieved by monitoring the current (A) due to reducing the enzymatic effect. The concentration range was found as 0–10,000 ng, where the detection limit was 5 ng/L and the quantification limit was 30 ng/L. An indirect competitive ELISA was exhibited in Figure 1d [55], competition occurred between aflatoxin B1-bovine serum albumin (AFB1–BSA) and free AFB1 (in peanut sample and standard) for the binding site of a fixed amount of anti-AFB1 antibody on the multi-walled carbon nanotubes/chitosan/screen-printed carbon electrode (MWCNTs/CS/SPCE). Figure 1e showed the effect of various blocking agents on background reading by eight percent skimmed milk, one percent BSA, casein, protein-free, and superblock [55].

### 4.3. Nucleic Acid-Based Bio-Recognition Materials

Compared with enzyme and antibody, nucleic acid (DNA or RNA)-based electrochemical biosensor was reported later, but their various applications are increased exponentially due to their multiple advantages. Their conformation is more robust than antibodies or enzymes. They can be entrapped in the biosensor assembly and bind with a wide range of specific targets with elevated affinity and sensitivity [65]. The interaction between immobilized nucleic acid and the analyte can change structures and electrochemical properties. One of the analytes are aptamers, which have artificial functional single-stranded DNA or RNA structures that can bind various target molecules, such as amino acids, small molecules, proteins, and cells, with high specificity affinity [66]. Aptamers can be obtained through an in the vitro selection procedure, followed by the classical methodology of systematic evolution of ligands by exponential enrichment (SELEX). Tuerk and Gold first proposed it in 1990 [67]. Figure 1f showed a highly sensitive impedimetric aptasensor for the selective detection of acetamiprid and atrazine [57].

Many nucleic acid-based electrochemical biosensor configurations have been extensively studied in gene analysis, clinical diagnostics, and environmental monitoring due to their fast, low-cost, sensitive, and selective responses to numerous analytes [68]. The most crucial step in preparing the nucleic acid-based electrochemical biosensor is the surface immobilization of the oligonucleotide strands. A terminal modification (sulfhydryl and amino groups) is the most common method to immobilise nucleic acid. Its most significant advantage lies in efficiently achieving directional and stable fixation. It is easy to prepare DNA arrays and realize high-throughput determination combined with lab-on-a-chip technology. The electrochemical detection of nucleic acid can be divided into direct and indirect methods. The electroactivity of oligonucleotide strands can be changed in the direct methods, which also changes the interfacial properties of the oligonucleotide strands-modified electrode in terms of conductivity, capacitance, or impedance. The indirect methods depend on the usage of electrochemical active nucleic acid labels [69] or intercalators [70] (e.g., methylene blue) [71].

Nucleic acids have been most widely used in metal ion detection, mainly consisting of the following four types: metal ion-specific DNAs, aptamers, DNAzymes, and guanine (G)-rich oligonucleotides, which can be related to G-quadruplexes [72]. Heavy metal ions can generate the partial disordering of oligonucleotide strands and reduce base stacking and base pairing after forming a metal–base complex. DPV studied the evaluation of the interaction of Pb^2+^, Cd^2+^, Ni^2+^, and Pd^2+^ with dsDNA, including hydrogen bonding cleavage, double helix conformation, and oxidative damage to DNA bases at GCE. [73]. Hg^2+^ can combine with two thymine bases (T) and mediate T–T mismatch to form a stable T–Hg^2+^–T structure which is more durable than the natural adenine–thymine (A–T) base pair with a binding constant close to 10^6^ M^−1^ [74]. Ag^+^ can selectively interact with cytosine (C)-rich oligonucleotide strands to form C–Ag^+^–C mismatch [75]. These impressive mismatches belong to coordination bonds, and on these principles, significant efforts have been made for high selectivity and sensitivity to determine Hg^2+^ [40,76] and Ag^+^ [77,78]. As for the detection of Pb^2+^ ion, the G-rich DNA sequences are widely used due to their ability to fold to form a most compact G-quadruplex structure, especially in the presence of Pb^2+^ ion [79]. For example, the simultaneous detection and determination of mercury (II) and lead (II) ions were implemented by Wang et al. [80]. The biosensor functionality was improved by placing the amino-modified reduced graphene oxide (NH_2_-rGO) nanofilm on a gold electrode as an excellent anchorage for the DNAzyme and the DNA strands. The presence of target ions could be recognized through the difference in charge-transfer resistance values before and after DNA interactions with Hg^2+^ and Pb^2+^ ions.

### 4.4. Whole Cell-Based Bio-Recognition Materials 

Whole cells or microorganisms used for environmental biosensing can be classified as bacteria, yeasts, and fewer algae. The whole cell-based biosensor combines cells and transducers, generating a measurable electrical signal against the specific or target analytes [81]. In recent years, the whole cell has become an excellent alternative to the traditional bio-recognition elements due to their easy cultivation and manipulation, hosting many enzymes to catalyze reactions and good compatibility with various types of transducers. Substantial efforts have been made, from commercial to well-characterized cells with robust and specific enzymatic properties [82]. Moreover, they can give information on the pollutants’ bioavailability and toxicity toward eukaryotic or prokaryotic cells [83].

Whole cells played an essential role in detecting heavy metal ions as the carrier to adsorb, precipitate or metabolize heavy metal ions. The whole cell was integrated into biosensors for low cost, low toxicity, high adsorption, and feasible fabrication based on the complexation, ion exchange, and physical adsorption between the whole cell and metal ions. Alpat et al., used green microalgae (Tetraselmis chuii) for the biosorption, preconcentration, and determination of Cu^2+^ in an easy, inexpensive, sensitive, and effective way [84]. The working electrode was fabricated by mixing green microalgae and carbon paste. Different pulse cathode differential voltammetry showed good linearity in the range of 5.0 × 10^−8^–1.0 × 10^−6^ M with the L.O.D. of 4.6 × 10^−10^ mol L^−1^. A Phormidium sp. modified voltammetric sensor for Pb^2+^ detection from aqueous solutions was also developed. Possible functional groups involved in Pb^2+^ accumulation were carboxyl, sulphoxide, and alcoholic groups. The developed microbial biosensor’s analytical properties and selectivity were investigated comprehensively, with a detection limit of 2.5 × 10^−8^ M [85]. 

The oxygen consumption estimates the biological oxygen demand (BOD) during the biodegradation with the aerobic whole cells as the catalysts. It is known that the biosensors need biorecognition elements with minimal selectivity and high activity of bio-oxidation for a wide range of organics, which ensures their application in the practical water samples containing water nutrients and complex organics. Xia’s group developed the fast detection method of BOD by selecting the Bacillus subtilis as a biorecognition element for its resistance in extreme conditions. They created a single-microbial-layered structure on the gold surface where the Bacillus subtilis bonded covalently. However, the conductivity was low due to the microbial electrode, and the biocompatibility was also poor [86]. In a second study, they improved the performance by creating the rough electrode surface with the microbial layer, and the carboxyl graphene and Au nanoparticles’ electrodeposition was used for creating this roughness [87]. In a third study, they used magnetite-functionalized Bacillus subtilis as the element of this BOD microsensor that can be regenerated and immobilized on an ultramicroelectrode array (UMEA). Modification and regeneration of the electrodes array are controlled magnetically. The assay can be performed in a short time (5 min) with vastly improved sensitivity. The calibration plot is linear in 2–15 mg·L^−1^. The developed biosensor was also applied successfully to determine BOD in spiked water samples [88]. Khor et al., constructed a two-electrode sensor system using calcium alginate to immobilize microorganisms for BO detection [89]. Ferrocyanide can dissolve in the carrier solution and be fixed into the membrane to participate in a biochemical reaction. Ultramicroelectrode has the advantages of small size, fast diffusion and mass transfer, and a fast, stable state. In the combination of optimized microbial-sensitive film thickness, the rapid detection of BOD is realized.

**Table 2 biosensors-12-00551-t002:** Summary of recognition elements, analyte, electrode, type of transducers, the limit of detection, and the response of various electrochemical biosensors.

Recognition Element	Analyte/Pollutant	Electrode/Sensing Material	Type of Transducers	Limit of Detection	Response Time	Response Range	References
**An enzyme (HRP)**	phenol	electrochemically reduced graphene oxide/glass carbon electrode	differential pulse voltammetry	2.19 μM	-	3.0–100.0 μM	[90]
**Enzyme (BChE)**	paraoxon	Prussian Blue Nanoparticles/screen-printed electrodes	amperometric	1 μg L^−1^	10 min	2.0–10 μg L^−1^	[91]
**Enzyme (AChE)**	Chlorpyrifos	ZrO_2_/RGO/ITO glass electrode	amperometric	100 fM	-	0.1–1000 pM	[92]
**An enzyme (tyrosinase)**	PhOH	ZnO Nanoparticles/screen-printed carbon electrodes	amperometric	19.8 nM	<10 s	0.1–14 μM	[93]
**Antibodies (anti-Microcystin-leucine arginine)**	Microcystin-leucine arginine	cysteamine/gold electrode	electrochemical impedance spectroscopy	570 pg L^−1^	-	3.3 × 10^−4^–10^−7^ g L^−1^	[94]
**Antibodies (anti-alkylphenols)**	4-nonylphenol	single-walled carbon nanotubes/gold electrode	field effect transistors	5 µg L^−1^	-	5–500 µg L^−1^	[21]
**Aptamers**	atrazine	platinum nanoparticles microwires	electrochemical impedance spectroscopy	10 pM	10 min	100 pM–1 μM	[57]
**Aptamers**	Vibrio alginolyticus	magnetic beads with a solid-contact polycation-sensitive membrane	potentiometric	10 CFU mL^−1^	1 min	10–100 CFU mL^−1^	[95]
**Whole cell (Shewanella cells)**	riboflavin	Shewanella oneidensis MR-1	amperometric	0.85 ± 0.09 nM	-	2–100 nM	[96]

## 5. Type of Transducers 

A transducer is considered a significant part of the biosensor. It can convert the physical change of the surroundings to suitable electrical signals, which a reaction could cause. Biological or bio-recognition elements can be associated with or integrated with a transducer. The bio-recognition elements can be incorporated with chemical and physical bounding on the surface of the transducer, which also depends on the immobilization methods [97]. Significant electrochemical transducers are available, such as amperometric or voltammetric, impedimetric, capacitive, potentiometric, and ion-selective field-effect transistors. The following sections summarise these transducers for water quality detections in various applications. As is shown in Table 3, various types of transducers and characteristics were summarized.

### 5.1. Voltammetric/Amperometric Biosensors

The method is based on the current detection technique, either by ramping up the working electrode’s potential at a given rate or keeping the potential constant compared to the reference electrode. The system’s response would be observed in both methods [65]. An amperometric biosensor (Figure 2) is based on the current generated from any electrochemical oxidation and reduction mechanisms of any electroactive species. It consists of a three-electrode system where a time-dependent excited potential is applied to the working electrode-changing the potential which is also relative to the fixed potential to the reference electrode. A current flows between the working electrode and the auxiliary electrode (nA to µA), where it is correlated with a bulk concentration of the electroactive species or the construction and expenditure rate within the adjoining biocatalytic layer. Platinum wire can be used as auxiliary electrode and an Ag/AgCl electrode can be used as reference electrode.

In 1956, Leland C. Clark introduced the oxygen probe, the simplest form of an amperometric biosensor. The oxygen probe measures the dissolved oxygen during the electrochemical reduction of oxygen. The associated electrolyte current is considered a response signal. This method can suit affinity sensors, which provide the electrochemically active compound as the recognition material and electrochemical labelling. Some electrochemically active nucleobases are included in the nucleic acid structure and are used for monitoring the recognition of hybridization. A new design [99] of biosensor strips was integrated with a conducting copper track and a graphite–epoxy composite for pesticide analysis. It was applied by screen-printing, and the enzyme (AChE or BChE) was immobilized manually by crosslinking with glutaraldehyde. Micaela Badea et al. [100] have reported modified platinum electrodes with a cellulose acetate membrane to fabricate rapid amperometric detection of nitrites and nitrates in water. The developed method is simple, fast, and does not need an extra reagent for nitrite detection.

Biagiotti, Vanessa, et al. [101] have reported a platinum electrode modified by electropolymerized films and polymer nanotubule nets. They tried several analytical parameters; among them, poly(1,3-DAB) film showed the best performance for nitrite detection in drinking water. The electrode was characterized electrochemically by cyclic voltammetry and amperometry coupled to flow injection analysis (FIA). It has shown the linear range of concentration (10–1000 μM), LOD (2 μM), and good reproducibility (R.S.D.%: 0.4). Stoytcheva, Margarita, et al. [102] undertook a work to determine the enzymatic phenols by developing polymer film formation on the working electrode. Pan, Yanhui, et al. [103] developed an electrochemical biosensor that was constructed by nitrogen-doped graphene nanoribbons (NGNRs) and ionic liquid (IL). The molecularly imprinted polymer (MIP) was used to develop the composite film to determine 4-nonyl-phenol (NP), and the determination of concentration range was 0.04–6 μM. They obtained satisfactory results from real samples with high sensitivity, selectivity, and stability.

**Table 3 biosensors-12-00551-t003:** Various types of transducers and characteristics.

Characteristics	Bio-Recognition Element	Detection Range	LOD	Response Time	Application	Ref
indium tin oxide (ITO) nanoparticles, hexaammineruthenium (III) chloride (RUT), and chitosan (CH) modified glassy carbon electrode (GCE)	horseradish peroxidase (HRP) enzyme	0.009–0.301 M (Pb^2+^), 0.011–0.368 M (Ni^2+^), and 0.008–0.372 M (Cd^2+^).	8 nM (Pb^2+^), 3 nM (Ni^2+^), and 1 nM (Cd^2+^)	10 s	Heavy metal detection in water, with good selectivity, stability, and reproducibility	[104]
glassy carbon electrode with gold nanoparticles	Pb (II)-DNAzyme	1 pM–1000 nM	0.42 pM	-	Heavy metal detection in water. High sensitivity, excellent specificity, good stability and acceptable reproducibility	[105]
glassy carbon electrode	*E. coli* cells immobilization using bovine serum albumin (BSA)	4.99 × 10^−10^ to 4.99 × 10^−3^ mol/L for mercury, 8.89 × 10^−10^ mol/L to 8.89 × 10^−3^ mol/L for cadmium, and 15.29 × 10^−10^ mol/L to 15.29 × 10^−3^ mol/L for zinc.	5.58 × 10^(−11)^ mol/L for mercury ion, 5.10 × 10^(−10)^ mol/L for cadmium ion, and 1.38 × 10^(−9)^ mol/L for zinc ion.	-	Heavy metal detection in water and lowcost and easy availability	[106]
glassy carbon electrode (GCE) modified with multiwalled carbon nanotubes (MWCNT)	choline oxidase enzyme	0.1 to 1.0 nM (Pb^2+^)	0.04 nM	5 min	Heavy metal detection in tap water	[107]
Pt/CeO_2_/urease electrode	ceria (CeO_2_) nano-interface	0.5–2.2 (Pb^2+^) and 0.02–0.8 μM (Hg^2+^)	0.019 ± 0.001 μM (Pb^2+^) and 0.018 ± 0.003 μM (Hg^2+^)	<1 s	Heavy metal detection in river water and good repeatability and reproducibility	[108]

### 5.2. Impedimetric Biosensors

An impedimetric biosensor (Figure 2) was fabricated by immobilizing the bio recognition elements onto the surface of the electrode. Different bio-recognition elements can detect nutrients, heavy metals, or waterborne pathogens. The targeted analyte can be measured through the output of an electrical impedance signal made proportional to activity of the analyte. It is a two-electrode system where the alternating voltage can be applied with a few to 100 mV amplitude. The impedance (Z), or the components of resistance (R) and capacitance (C), can be changed due to the behaviour of the material. The applied voltage frequency can be scanned over various frequencies to get the corresponding impedance and characterize the sensor for specific material. The equivalent circuit parameters are also used for impedance spectra for characterization purposes. For developing an impedimetric biosensor, the prerequisite condition is the reproducible ability of the immobilizing bio-recognition molecules onto the sensor surface with the possession of their biological activity [109].

The impedance spectrum can be displayed in Nyquist or Bode plots. The plot is a semicircle region lying on the axis, followed by a straight line. Usually, electrochemical impedance spectroscopy (EIS) is used to investigate the properties of bio-recognition events at the modified surface.

An impedimetric biosensor was reported [110] with highly conductive tantalum silicide (TaSi_2_) to detect and quantify *E. coli* O157:H7 in drinking water. The developed biosensor shows a linear response with a concentration of 101–105 CFU mL^−1^ and a sensitivity of 2.6 ± 0.2 kΩ. It can avoid interference which also confirms the excellent selectivity. The developed biosensor can be used multiple times with good repeatability. Hnaien, M. et al. have reported [111] a bacterial impedimetric biosensor for trichloroethylene (T.C.E.) detection in drinking water. Gold microelectrodes were used with single-wall carbon nanotubes, further linking with anti-Pseudomonas antibodies. It also showed a good linear response with the T.C.E. concentration up to 150 μg L^−1^ and a low L.O.D. (20 μg L^−1^). It also showed excellent stability and recovery in real sample water. Lin, Zhenzhen et al. [112] have reported a biosensor for simultaneous detection of metal ions, such as Pb^2+^, Ag^+^, and Hg^2+^ in lake water. The DNA-based bio-recognition element was immobilized on the working gold electrodes. The developed biosensor had high sensitivity and selectivity, which were evaluated using the charge transfer resistance (RCT) difference before and after the immobilized DNA interactions with Pb^2+^, Ag^+^, and Hg^2+^. Madianos, L. et al. [113] developed a biosensor to detect acetamiprid and atrazine (pesticides) in natural water. The e-beam lithography technique deposited platinum nanoparticles (Pt NPs) between the interdigitated electrodes (IDEs) to create a bridge structure. The aptamer was chemically used to functionalize the Pt N.P.s on the sensing surface. The developed biosensor was highly sensitive and selective and also showed excellent linear response in the range of 10 pM to 100 nM for acetamiprid and 100 pM to 1 μM for atrazine.

### 5.3. Capacitive Biosensors

Capacitive biosensors consider be the group of affinity biosensors that operate by the direct binding between the surface of the sensor surface and the target molecule. It measures the variations in the dielectric properties and/or the thickness of the dielectric layer at the electrolyte/electrode interface location. A conventional electrical plate capacitor contains two conductive metal plates with specific dielectric properties separated by a certain distance. The following relations can express the:(1)$$C=\frac{€A}{d}$$
where € is the permittivity of the dielectric material, *A* is the area of the plate, and *d* is the distance between them. Therefore, when there is a change in the properties of the materials, a change in capacitance can be measured by the above equation. The second type of capacitive biosensor depends on the theory of electrical double-layer. The electrodes submerged in an electrolyte solution can resemble a capacitor for storing charge where an insulating layer covers the surface. The specific biorecognition element can be immobilized on top of this layer. The solvated ions and water molecules create a capacitance near the electrode surface.

N. V. Beloglazova et al. [114] reported a capacitive biosensor to detect benzo(a)pyrene (BaP) in river water. MIPs and monoclonal antibodies (mAb) are used as recognition elements on the electrode. The sensor is validated in a contaminated water sample from different places in Ghent, Belgium. Graniczkowska et al. [115] reported the development of a capacitive biosensor to monitor an amphetamine as a trace amount in water samples. The gold sensing electrode is immobilized with MIPs for creating sensing elements. Samuel M. Mugo et al. [116] reported a pathogen imprinted polymer for detecting *Escherichia coli* in water. The conducting electrode is based on multi-walled carbon nanotubes (CNT), and nitrocellulose (CNC) films, which were integrated with polyaniline (PANI) doped phenylboronic acid (PBA). The proposed sensor used both the capacitive and impediometric method for detecting the *E. coli* with a rapid response of ≤5 µmin.

### 5.4. Conductometric Biosensors

Conductometric biosensors measure the conducting current between the electrodes and reference electrodes where the analyte or the medium plays a vital role. Usually, a differential measurement is performed between the working electrode with an enzyme and an identical reference electrode without an enzyme in a biosensor. The sensitivity of the sample amount is hampered by the parallel conductance of the target solution. The technique is significantly like conventional conductometers. An alternating current with the operating frequency is applied to the active electrodes to measure the potential. Conductance is measured by using both the current and voltage. Glucose, urea, creatinine acetaminophen, and phosphate are reported as different analytes to be determined using conductometric biosensors [117].

G. A. Zhalyak et al. [118] reported an alkaline phosphate-based conductometric biosensor for assessing the heavy metal ions in water. Gold-based electrode and residual enzyme activity measured in tris-nitrate buffer without metal preincubation. Various metal toxicity can be measured in the range as follows: Cd^2+^ > Co^2+^ > Zn^2+^ > Ni^2+^ > Pb^2+^. A similar method (Figure 3) is reported [119] to identify the heavy metals in water. The alkaline phosphate activity (APA) was collected from cyanobacterium to immobilize directly on the substrate by physical absorption. The response time was 12 s. Other works [120,121,122] are also reported for heavy metal detection in water.

In this work [125], the proposed biosensor was developed to determine the organic matter in water by immobilizing the enzyme bilayer with bovine serum albumin in glutaraldehyde vapour. It can detect the protein as a biomarker in water to identify urban pollution. The proposed method shows excellent sensitivity, reproducibility, and a long lifetime. C. Chouteau et al. [126] reported the whole cell *Chlorella vulgaris* microalgae as a bioreceptor on the interdigitated conductometric electrodes for detecting the toxic compounds in aquatic habitats. N. Kolahchi et al., proposed a fast, sensitive miniaturized conductometric biosensor for determining the phenol in water. *Pseudomonas* sp. (GSN23) bacteria were immobilized on the gold interdigitated microelectrodes to create the sensor assembly to determine phenol in river water.

### 5.5. Potentiometric Biosensors

A potentiometric biosensor works on the principle of potential difference between the working electrode and the reference electrode. The measured analytes are not consumed in the same way as in the amperometric biosensor. In this biosensing method, two electrodes galvanic cells immersed in the electrolyte solution generate the electromotive force (e.m.f.) measured by a high impedance voltmeter [127]. One electrode is used as a working electrode, and another is used as a reference electrode. The e.m.f. value is determined by the potential difference between the two electrodes. The analyte’s concentration and the potential difference is measured by the Nernst equation [128], which is explained as follows:(2)$$E_{cell}=E_{0}-\frac{RT}{zF}lnQ$$
where *E_cell_* is the e.m.f., *E*_0_ is the potential of the standard electrode, *R* is the gas constant, *T* is the temperature in Kelvin, *z* is no of charge of the electrode reaction, *F* is the Faraday constant, and *Q* is ion concentration ratio of the anode to cathode.

Huang, Mei Rong et al. [129] have reported a membrane based on semi-conducting poly(phenylenediamine) microparticles for Pb^2+^ detection in natural water. The electrode is selective towards the Pb^2+^ with the concentration range 3.16 × 10^−6^ to 0.0316 M with a high sensitivity displaying a near-Nernstian slope of 29.8 mV decade^−1^. The proposed electrode showed a long lifetime of 5 months, where the short response time was 14 s. Thayyath S. Anirudhan, and S. Alexander [130] have developed a modified multiwalled carbon nanotube (MWCNT) based imprinting polymer for the determination of pesticide 2,4-D (2,4-dichlorophenoxyacetic acid) in natural water. The sensor responds in the range of 1 × 10^−9^–1 × 10^−5^ M, where the detection limit is 1.2 × 10^−9^ M. The developed sensor is stable and can be reusable many times in the first 3 months. Mashhadizadeh, Mohammad Hossein et al. [131] have reported a newly modified carbon electrode to determine the Cu^2+^ with the presence of other interfering ions. The proposed potentiometric sensor showed a Nernstian slope of 30 (±0.5) mV/decade over a concentration range from 1.0 × 10^−8^–1.0 × 10^−2^ mol L^−1^. The LOD was 7.0 × 10^−9^ mol L^−1^ and the response time was 30 s, which can be used for at least 3 months without sacrificing any quality of the sensor’s response without any considerable divergence in responses.

### 5.6. Ion-Selective Field-Effect Transistors (ISFET) Based Biosensors

The last few years saw a significant change in ion-sensitive field-effect transistor (ISFET)-based devices, a principle initially proposed in the 1970s by Bergveld et al. [132]. This revolutionized the technology at a nanoscale level. These types of prototypes formed using silicon nanowire FETs (SiNWFETs) have been used for a wide range of applications, including pH sensing [133,134,135], chemical [136,137,138,139], and label-free biosensing [133,140,141,142,143] applications. The downscaling of these devices has been done by determining the kinetic studies on receptor binding [144] and intracellular recordings of action potentials [145]. The working mechanism is based on the adsorption of charged species on the sensing surface, causing variation in the surface potential and, thus, the current in the FET channel. These devices provide an additional attribute over the conventional ion-selective electrodes by transforming the high-impedance input signal into a low-impedance output signal. The probability of downscaling the dimension and conjugation them with conditioning circuits to detect multifunctional parameters [146] highlights the ability of SiNWFETs to operate as low-cost, efficient, and robust devices.

In one of the examples, the selectivity of the sensing surface is induced to deduce species other than protons. This was carried out to achieve high sensitivity by absorbing the target analyte. The covalent bonding of the linked molecules to perform chemical anchoring has been more than a viable method. The linker binding sites of the self-assembled monolayers (SAMs) are situated in the proximity of the FET surface at a higher density. The topic of SAMs has extensively been studied [49]. In ISFET systems, a technique related to self-assembly of silane monolayers has been used to alter the surfaces of the oxides [133,136,147].

Ion-sensitive field-effect transistors that use silicon nanowires have high dielectric constant gate oxide layers formed with Al_2_O_3_ or H_2_O_2_. These devices exhibit responses that are sensitive to pH variations and ions present in the electrolyte due to the presence of hydroxyl groups. The complexity of the oxide surface due to its intrinsic non-selective nature makes it challenging to sense ionic species other than protons. The modification of the individual nanowires has been done with thin gold films to increase the specificity via functionalization. It has also been shown that the sodium ion (Na^+^) detection using SAM of thiol-modified crown ethers has been done, where a response of ≈−44 mV per decade was achieved for the sodium ions in a NaCl solution. The testing process was carried out in the presence of various ions like protons (H^+^), potassium (K^+^), and chloride (Cl^−^) ions, where the voltage difference between the gold-coated nanowire functionalized by the SAM (active) and a gold-coated nanowire was determined. It was seen that the functional SAM was unable to obtain any output from the bare gold-coated nanowire concerning pH and background ionic species. This response increases the credibility of gold in comparison to oxide surfaces when the devices are used for differential measurements.

## 6. Signal Amplification Strategy

One of the critical capabilities of a biosensor lies in its enhanced performance in terms of the morphological, structural, and electrochemical characteristics of the considered nanostructured material. In normal terms, the sensors are characterized using X-ray diffraction, confocal microscopy, transmission electron microscopy, and voltammetric techniques. The hybrid prototypes used for biosensing applications constitute combined attributes that have generated high sensitivity, selectivity, and rapid and stable response during the detection water pollutants. The synergy observed between the processed materials improved the electrochemical activity, stability of the immobilization of bioreceptor, electron transfer rate and surface area of the electrodes, thus obtaining a high magnitude of the peak current during the detection of different analytes as a typical signal amplification strategy.

The use of metallic nanoparticles with a large electroactive surface area has been employed for electrochemical biosensing applications due to their high electrical conductivity, catalytic properties, fast electron transfer, biocompatible nature, and easy incorporation with different kinds of nanomaterials. Zeinab et al., showcased the use of an ultrasensitive electrochemical aptasensor for quantitative detection of bisphenol A (B.P.A.) via signal amplification strategy

The use of gold-platinum nanoparticles (Au-PtNPs) was carried out by electrodepositing them on acid-oxidized carbon nanotubes (CNTs-COOH)-modified glassy carbon (GC) electrodes. Then, acriflavine was immobilized by covalent bonds at the surface to capture BPA-aptamer by forming phosphoramidite bonds. The aptamer’s conformational change occurred once the B.P.A. interacted with the aptamer. Thus, the retardation was carried out for the interfacial electron transfer of acriflavine as a probe. The LOD for this technique was calculated to be as low as 0.035 pM, which resulted from high-density Au-PtNPs and superior electron transfer of acriflavine. The resulting aptasensor also exhibited reasonable specificity, stability, and reproducibility.

Recently, the application of conducting polymer-based materials were made in biosensors for two particular areas, including enhancing the affinity of these sensors as backbone or side groups and using them as immobilization matrices for the bioreceptors with high electrical conductivity and fast electron transfer [44]. Disposable nitrate biosensors were used as nanoarrays to detect nitrate reductase as a target analyze. Bio-recognition element was immobilized within a poly (3, 4-ethylenedioxythiophene) (PEDOT) matrix to produce a quantifiable amperometric response. Superior analytical performance and quick fabrication process, and easy operating principle were obtained with this PEDOT/nitrate reductase nanowire sensor [148].

Dendrimers are synthetic three-dimensional macromolecule polymers with well-defined, highly branched, globular-shaped molecular structures [149]. Poly (propylene imine) dendrimer PPI has been popularized for biosensing applications due to its high biocompatibility and compatibility with host–guest chemistry. Due to the disastrous effects of cholera infection resulting in watery diarrhoea with severe dehydration and death, Tshikalaha et al., developed biosensors to detect cholera toxins in the water. The sensors are operated by co-electrodepositing PPI dendrimer and AuNPs on glassy carbon electrodes. The probe of the anti-cholera toxin was dropped on GCE/PPI/AuNPs and finally blocked with B.S.A. to reduce nonspecific binding[150]. A LOD of 7.2 × 10^−13^ and 4.2 × 10^−13^ g mL^−1^ were obtained from SWV and EIS analysis.

Another significant strength in improving electrochemical signals is the core-shell or core-satellite nanostructures. These nanomaterials can be loaded to the core nanoparticle via surface functionalization in chemical ways. For example, heterogeneous magnetic nanoparticles [151] and Mesoporous silica [152] have attracted increasing attention due to their easy magnetic separation with labelled bio-receptors and easy encapsulation of enormous materials in their structural pores, respectively. Ultrasensitive electrochemical biosensors used to detect Ag^+^ ions were constructed using magnetic Fe_3_O_4_@gold core–shell nanoparticles (Fe_3_O_4_@Au NPs) for labelling with H.C.R. product and enrichment on the surface of the magnetic gold electrodes [153]. The prototypes showed high selectivity due to their duplex-like DNA. scaffolds structure with specific C–Ag^+^–C base pairing. They also had attributes like high sensitivity, low LOD of 0.5 fM and a wide dynamic range of 1 fM–100 pM. Marcos et al. [154] reported a new hybrid nanomaterial platform that included MWCNT and haemoglobin-functionalized mesoporous silica particles with highly sensitive quantification of nitrite and trichloroacetic acid as processed materials. The efficient electron transfer between haemoglobin and the electrode surface can be attributed to certain factors, including the high surface area and protein loading capacity of the mesoporous silica nanoparticles as well as the increased surface area and catalytic properties of MWCNTs.

In regards to the porous materials, metal–organic frameworks (MOFs) are another interesting class of porous crystalline inorganic–organic hybrid materials, as Fe(III)-based MOF (Fe-MOF) was reported [155] to have an excellent stability and redox activity when used as the prime electrode materials [156]. As is shown in Figure 4, a core–shell nanostructured Fe(III)-based metal–organic framework (Fe-MOF) and mesoporous Fe_3_O_4_@C nanocapsules (denoted as Fe-MOF@mFe_3_O_4_@mC)-based aptasensor was constructed [155]. The EIS was used for detecting the responses, where an advantage of the conformational transition interaction took place. This phenomenon was caused due to two factors: the formation of the G-quadruplex between a single-stranded aptamer and a highly heavy metal ion of Fe-MOF. The proposed aptasensor showcased a decent linear relationship with the logarithm of heavy metals and a low LOD of 2.27 and 6.73 pM toward the detection of metallic ions of Pb^2+^ and As^3+^, respectively.

## 7. In Situ Monitoring System

The recent development of various biosensors recommends excellent potential for monitoring water quality in the other treatment water recourses due to their simple, compact design, dispensability, and cheapness. In situ can be considered as online monitoring and offline or portable monitoring. Online monitoring defines as real-time in situ measurements of any sampling for analysis and provides on-field sampling data compared to conventional methods. It is incredibly challenging to monitor water contaminants, primarily chemical pollutants, in online monitoring. It is a more flexible approach and can be conducted from remote locations. An online monitoring system can be constructed utilizing a wireless sensor network (WSN) or an Internet of Things (IoT)-enabled network [25,157,158]. Simultaneous data collections, higher detection, easy monitoring, and sufficient data are the significant advantages of constructing the WSN network for monitoring purposes. Low power consumption and energy harvesting options are essential for developing an online monitoring network.

Pasternak et al., reported [136] a biological oxygen demand (BOD) biosensor, which was self-powered and autonomous for water quality measurement. The energy harvesting system, data logger, and sensing unit were developed continuously to monitor the sample in water (Figure 5). This biosensor can detect urine contamination in water, and the system can run autonomously for five months. 

Quek et al. [160] reported an assimilable organic carbon (AOC) based amperometric biosensor for detecting marine microbial fuel cell (MFC) in marine water, where the system was tested for 36 days. The response time, the reproducibility of the signal, and recovery time were good, which are essential for developing an online monitoring system. Bio-recognition elements play a crucial role in developing robust biosensors, which could be helpful for online monitoring. Among the many bio-recognition features, the enzymatic biosensors method is used widely for electrochemical detection, as they have high sensitivity for distinguishing the target elements from interference elements [161]. However, they are a costly method, have an increased duration of immobilization procedure, and have poor durability and stability, which is ascribed to the loss of enzyme activity during the onsite monitoring [82]. Therefore, MFC biosensors are widely used for various target analytes with an extensive range of cells [162]. They have mainly been installed to monitor water quality, but very few commercial prototypes are available for monitoring water toxicity. They can survive under harsh conditions, such as high and low pH, unusual temperature, and salinity [161].

Figure 6 has shown the portable electrochemical EIS based system for monitoring samples from water. Figure 7 shows the schematic block diagram of a standard electrochemical biosensor monitoring system. The sensing parameters would be capacitance, impedance, current, or voltage based on the characteristics of the electrochemical biosensor. The impedance analyzer relates to the sensor to collect the sensor data. It also provides sufficient energy to the sensor. The microcontroller unit manages all the sensing data, sends the data to the cloud server through the base station or the internet and manages the operating condition of the complete sensing system. The energy harvesting unit connects with the power management unit to supply continuous energy to the sensing unit. The microcontroller unit also connects with the wireless communication module, another crucial module for developing an online monitoring system. Different wireless communication modules are available, such as Bluetooth low energy, low power wide area network (LPWAN) wireless modules, SigFox modules, WiFi modules, and ZigBee modules. The modules are characterized based on their bandwidth, data transmission capability, power consumption, and communication range. The communication module solely depends on the installation duration of the network, the number of sensing systems, the content of coverage regions of the networks, and the application.

## 8. Challenges and Future Work

Although much work has been carried out to detect water quality using electrochemical biosensors, some bottlenecks still need to be addressed. Some challenges in microbial biosensors’ detection process are low recognition limits, limited specificity, and high contamination. It also has limitations in mass transmission due to the subsequently limited penetration of substrates and products throughout the cells [163]. These bottlenecks primarily exist and limit the sensitivity during these biosensors’ real-time application. These prototypes can only detect certain microorganisms and limited chemicals present in water, thus making them unsuitable for the in situ monitoring of unanticipated shocks in wastewater [164,165]. Additionally, there is currently no verification on the immobilization of enzymes or microorganisms on the surfaces of biosensors during their deployment in harsh environments. They also have low durability, primarily when operated over several hours [165,166,167]. This makes these prototypes unsuitable for long-term operations during wastewater treatment. Other issues are the requirement of external power sources and additional dissolved oxygen (DO) that deters the exact conductivity and pH probes to measure various parameters [168,169]. These problems lead to the deterioration of their performance over a prolonged period. The real-time monitoring has also been challenging due to the delay in the response of these biosensors, thus hindering the timely action from overcoming the shock effects. For example, when anaerobic granule biosensors were used for the early alarm to detect copper and phenol in the wastewater [170], the delay (6–20 h) in the response time created problems in the practical application and of these prototypes.

Generally, electrochemical sensors have specific attributes like lower detection limit of detection than colourimetric and fluorescent sensors (pvalue ≪0.05, d-value >0.8) [171]. Some of the primary characteristics of the electrochemical biosensors are their compatibility with modern microfabrication technologies, low input power, roll-to-roll fabrication, and the independence of sample turbidity and colour [172].

Even though the fabricated sensors have been used for multifunctional applications, most focus on the detection criteria lying on the sampled genre like heavy metal ions. Rarely have the sensors been used to detect some multi-analytes like antibiotics, small molecules, and metal ions. Julius et al. [173] displayed the development and implementation of a cell-free in vitro transcription system that deploys RNA output sensors activated by ligand induction (ROSALIND) to detect specific contaminants based on aptamer transcription and fluorescent signal analysis. Importantly, easy storage and distribution can also be carried out with the ROSALIND system, thus making it easier to deploy. This assists in determining their capability to test municipal water supplies and demonstrate their use for monitoring water quality.

## 9. Conclusions

There is continuous fear about the risks caused by contaminations or pollutants to human health and marine ecosystems. However, standard analytical techniques are sensitive, accurate, laborious, expensive, and unsuitable for on-site monitoring with complex water sample pretreatment requirements before testing under the guidance of trained personnel. This review assessed the recent progress in developing electrochemical biosensors for water quality sensing applications over the current time. Many of them also have certain advantages over the other methods in detecting the aggregate outcomes of multiple pollutants in water samples. Although electrochemical biosensors have great potential and are very sensitive and cost-effective compared to the standard analytical methods, they still need to reduce their cost and response time performance compared to the other sensors. The research will likely continue by modifying the electrode surfaces and innovative biorecognition elements, using various nanomaterials, conducting polymers, etc., and improving the surface-by-surface modification techniques to enhance electrochemical biosensor sensitivity and selectivity and the quick response. Further integration with intelligent electronics and wireless technologies will significantly benefit the development of biosensors for remote sensing applications or in situ measurements. However, the stability of the electrochemical biosensor remains a challenge that needs additional research to explore and to extend its shelf life. Finally, this review outlines the current methods and technologies in electrochemical biosensors for water quality sensing applications. We think that this review article will be helpful for beginners and a helpful guide that will enhance the awareness of the role that electrochemical biosensors can play in protecting our environment and most valuable water resources.

## Figures and Tables

**Figure 1 biosensors-12-00551-f001:**
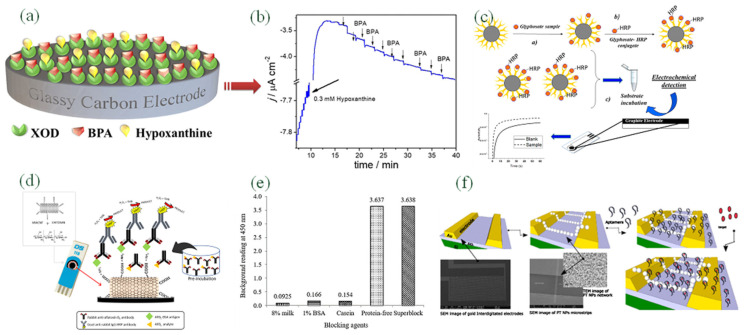
(**a**,**b**) Amperometric response of B.P.A. at XOD/GCE in the presence of 0.3 mM hypoxanthine (Reproduced with the permission from [53]); (**c**) Schematic of magnetic beads (MBs) for the analyte and its capturing technique on the electrode surface (Reproduced with the permission from [54]); (**d**) The complete schematic diagram of the nanomaterial-based immunosensor based on ELISA indirect competitive format (Reproduced with the permission from [55]); (**e**) Effect of various blocking agents on background reading by eight percent skimmed milk, one percent BSA, casein, protein-free, and superblock (Reproduced with the permission from [56]); (**f**) Schematic representation, SEM and EIS responses of the fabricated aptasensor (Reproduced with the permission from [57]).

**Figure 2 biosensors-12-00551-f002:**
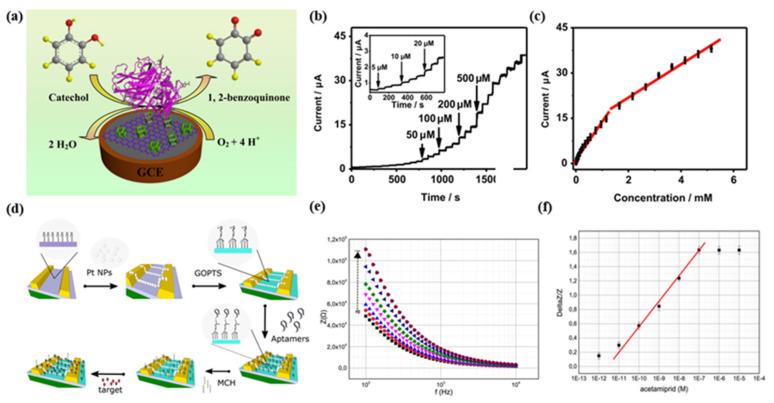
(**a**) Schematic design of the catalyzed oxidation on the catechol (analyte) electrode surface by laccase. (**b**) The proposed calibration curves of the catalytic currents vs. catechol (analyte) concentrations; and (**c**) calibration curve (Adapted from [98]) based on amperometric responses; (**d**) Schematic design of surface functionalization, where the Thiol-modified aptamers are bonded covalently and immobilized on the surfaces; (**e**) The Bode plots of the functionalized sensors; (**f**) The calibration curves are obtained for pesticides, such as acetamiprid (reproduced with the permission of [57]).

**Figure 3 biosensors-12-00551-f003:**
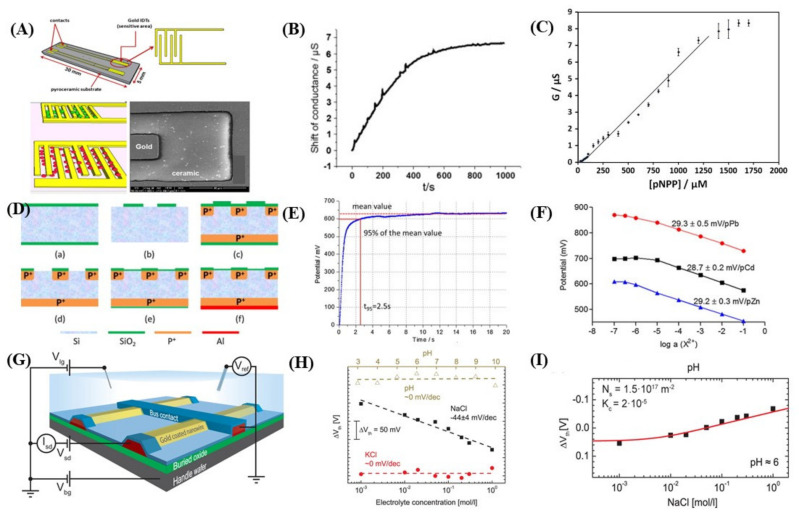
(**A**) Schematic of the microelectrodes with the gold electrode, the working electrode is immobilized with Spirulina cells, the reference electrode is immobilized with inhibited APA, which also includes Spirulina cells, and S.E.M. image of spirula cells with the gold electrodes of interdigitated transducers. (**B**) The real-time response of the conductometric transducer. (**C**) Standard calibration curve for the detection of the alkaline phosphatase activity (reproduced with the permission from [119]). (**D**) Fabrication process. (**E**) the response time of the sample solution. (**F**) The averaged calibration curve (reproduced with the permission from [123]). (**G**) Schematics of the measurement setup of the FET sensor. (**H**) Differential threshold voltage (ΔV_th_) measurement of the gold-coated NWs vs concentration of the electrolyte and pH. (**I**) Response of the ionic strength of the gold-coated NW fitted with a blended site-binding model for deprotonation, protonation, and Cl– adsorption (reproduced with the permission from [124]).

**Figure 4 biosensors-12-00551-f004:**
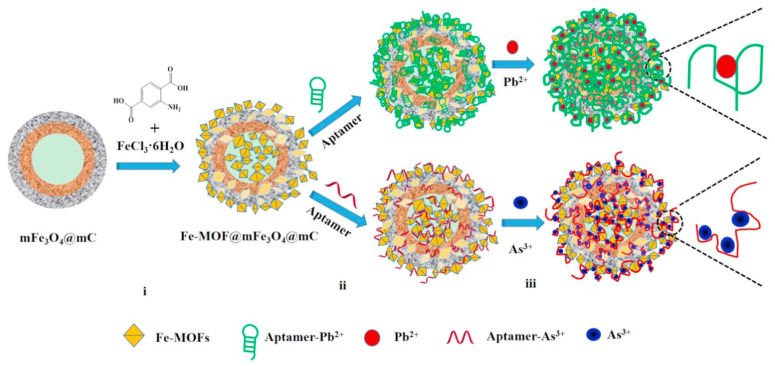
Schematic diagram of the preparation process of nanocomposite and its related aptasensor for detecting Pb^2+^ and As^3+^ via electrochemical techniques, including (**i**) the preparation of the nanocomposite, (**ii**) the immobilization, and (**iii**) the determination of the heavy metal ions, (reproduced with the permission from [155]).

**Figure 5 biosensors-12-00551-f005:**
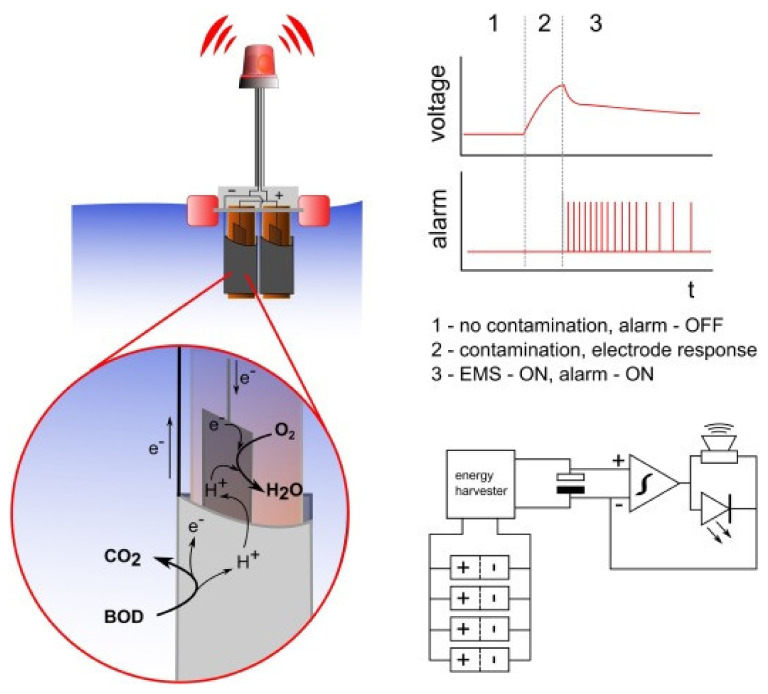
Schematic representation of the sensor’s proposed biosensor and operating principle. The block diagram of the system shows the energy harvester charging/discharging repeatedly (reproduced with the permission of [159]).

**Figure 6 biosensors-12-00551-f006:**
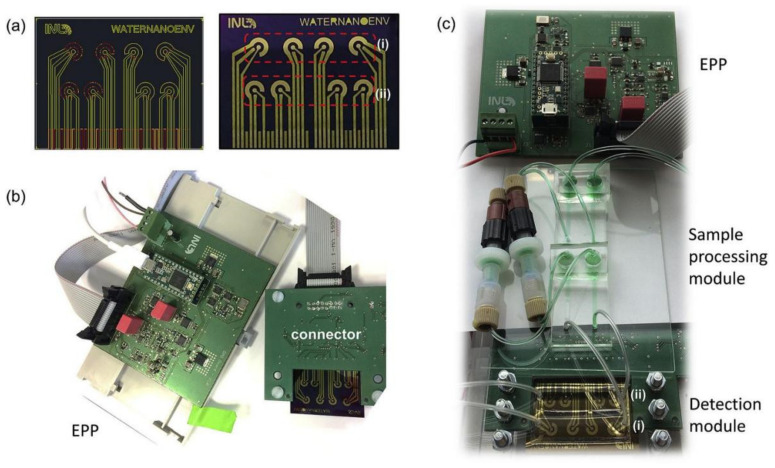
(**a**) Design and fabrication of electrochemical-cell-chips development; (**b**) electrochemical impedance portable platform for EIS measurements; and (**c**) complete portable system for automatic detection (reproduced with the permission of [94]).

**Figure 7 biosensors-12-00551-f007:**
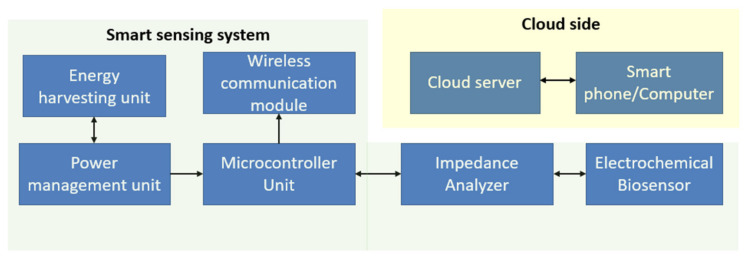
Schematic diagram of an online monitoring system.

**Table 1 biosensors-12-00551-t001:** Surface modification techniques of BRE in electrochemical biosensors.

Surface Modification Technique	Immobilization Site	Spatial Orientation	Accessibility	Advantage	Disadvantage	Ref.
Adsorption	random	random	low	simple and direct	low immobilization efficiency	[45,46]
Encapsulation in polymers or gel	random	random	low	abundant BRE	necessary surface treatment and low immobilization efficiency	[47]
Chemical crosslinking	random	random	low	simple and high stability	the strict control of conditions and nonspecific interaction	[48]
Self-assembled monolayers	active terminal	orientation	high	simple and controllable BRE density	possible nonspecific interaction	[49]
Covalent linking	terminal activation	orientation	high	high stability	necessary surface treatment and low immobilization efficiency	[50]
Affinity	biotinylated terminal	orientation	high	simple and high stability	necessary surface treatment and possible nonspecific interaction	[51]
Electrodeposition	random	random	high	Reliable, cost-effective, and easy fabrication and maintenance	possible nonspecific interaction	[52]

## Data Availability

Not applicable.

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
