# Peer review of "Recent Advancements in Electrochemical Biosensors for Monitoring the Water Quality"

_biosensors, 2022, doi:10.3390/bios12070551_

Round 1

Reviewer 1 Report

Hui and Col. Present a review article of  recent advancements in electrochemical biosensors for  monitoring the water quality. It is a good review, but I think that some aspects need further attention.

Line 100-102

" Biosensors have many advantages over the conventional lab-based method,  including low-cost, high sensitivity, selectivity, portability, and the capability  to monitor the complex toxicity of polluted water". In a practical level, biosensors still struggle with the obtention of high sensitivity and selectivity to compete with conventional lab-based method, please correct this sentence.

Line 113

"Bio-recognition", sometimes this word in a middle of a phrase is write it with capital letter . This minor mistakes are present in all the manuscript even with other words. Please correct mistakes.

Lines 164-169

Add references at the end of this phrase "Most pro teins usually achieve the best surface coverage on uncharged surfaces under  the neutral pH and functional ionic strength, using a specific 5−20 μg/mL concentration".

In the sections inside of " Type of Transducers", authors presented brief basic information, considering the I´m of the article " mainly scientists 129 who might be unaware of electroanalytical chemistry and biosensors ", is necessary deep a little more in information that allows the readers a mayor comprehension of how is carried out the quantification on these transducers. Also a scheme is suggested.

Author Response

Reviewer 1

Hui and Col. Present a review article of  recent advancements in electrochemical biosensors for  monitoring the water quality. It is a good review, but I think that some aspects need further attention.

Line 100-102

" Biosensors have many advantages over the conventional lab-based method,  including low-cost, high sensitivity, selectivity, portability, and the capability  to monitor the complex toxicity of polluted water". In a practical level, biosensors still struggle with the obtention of high sensitivity and selectivity to compete with conventional lab-based method, please correct this sentence.

Reply: Thanks for your insightful advice.  And we revised this sentence as follows " Biosensors have many advantages over the conventional lab-based method, including low-cost, portability, fast response time, less usage of reagents and the capability to continuous monitor the polluted water".

Line 113

"Bio-recognition", sometimes this word in a middle of a phrase is write it with capital letter . This minor mistakes are present in all the manuscript even with other words. Please correct mistakes.

Reply: "Bio-recognition" was corrected as "biorecognition". And we have gone through the manuscript carefully to correct all of them.

Lines 164-169

Add references at the end of this phrase "Most proteins usually achieve the best surface coverage on uncharged surfaces under  the neutral pH and functional ionic strength, using a specific 5−20 μg/mL concentration".

Reply: Thanks for the comment. We have added the reference after the end of this statement.

In the sections inside of " Type of Transducers", authors presented brief basic information, considering the I´m of the article " mainly scientists 129 who might be unaware of electroanalytical chemistry and biosensors ", is necessary deep a little more in information that allows the readers a mayor comprehension of how is carried out the quantification on these transducers. Also a scheme is suggested.

Reply: Thanks for the valuable suggestions. We have included the quantification of all the transducers methods in the relevant sections as follows:

“An Amperometric biosensor is based on the current generated from any electrochemical oxidation and reduction mechanisms of any electroactive species. It consists of a three-electrode system where a time dependent ex-cited potential is applied to the working electrode-changing the potential which is also relative to the fixed potential to the reference electrode. There is a current flow (nA to µA) between the working electrode and the auxiliary electrode where it is correlated with a bulk concentration of the electroactive species or the construction and expenditure rate within the ad-joining biocatalytic layer. Platinum wire can be used as auxiliary electrode and an Ag/AgCl electrode can be used as reference electrode.”

“An impedimetric biosensor is fabricated by immobilizing the bio recognition elements onto the surface of the electrode. Different bio-recognition elements can detect nutrients, heavy metals, or waterborne pathogens. The targeted analyte can be measured through the output of an electrical impedance signal made proportional to activity of the analyte. It is a two-electrode system where the alternating voltage can be applied with a few to 100 mV amplitude. The impedance (Z), or the components of resistance (R) and capacitance (C), can be changed due to the behaviour of the material. The applied voltage frequency can be scanned over various frequencies to get the corresponding impedance and characterize the sensor for specific material. The equivalent circuit parameters are also used for impedance spectra for characterization purposes. For developing an impedimetric biosensor, the prerequisite condition is the reproducible ability of the immobilizing bio-recognition molecules onto the sensor surface with the possession of their biological activity [111]. The impedance spectrum can be displayed in Nyquist or Bode plots. The plot is a semicircle region lying on the axis, followed by a straight line. Usually, Electrochemical Impedance Spectroscopy (EIS) is usually used to investigate the properties of bio-recognition events at the modified surface.”

“Capacitive biosensors consider to the group of affinity biosensors that operating by the direct binding between the surface of the sensor surface and the target molecule. It measures the variations in the dielectric properties and/or the thickness of the dielectric layer at the electrolyte/electrode interface location. A conventional electrical plate capacitor contains two conductive metal plates with specific dielectric properties separated by a certain distance. The following relations can express the:

Where € is the permittivity of the dielectric material, A is the area of the plate, and d is the distance between them. Therefore, when there is a change in the properties of the materials, a change in capacitance can be measured by the above equation. The second type of capacitive biosensor depends on the theory of electrical double-layer. The electrodes submerged in an electrolyte solution can resemble a capacitor for storing charge where an insulating layer covers the surface. The specific biorecognition element can be immobilized on top of this layer. The solvated ions and water molecules create a capacitance near the electrode surface.”

“Conductometric biosensors measure the conducting current between the electrodes and reference electrodes where the analyte or the medium plays a vital role. Usually, a differential measurement is performed between the working electrode with an enzyme and an identical reference electrode without an enzyme in a biosensor. The sensitivity of the sample amount is hampered by the parallel conductance of the target solution. The technique is significantly like conventional conductometers. Alternating current with the operating frequency is applied to the active electrodes to measure the potential. Conductance is measured by using both the current and voltage. Glucose, urea, creatinine acetaminophen, and phosphate are reported as different analytes to be determined using conductometric biosensors [118].”

“A potentiometric biosensor works on the principle of potential difference between the working electrode and reference electrode. The measured analytes are not consumed in the same way as in the amperometric biosensor. In this biosensing method, two electrodes galvanic cells immersed in the electrolyte solution generate the electromotive force (e.m.f.) measured by a high impedance voltmeter [128]. One electrode is used as a working electrode, and another is used as a reference electrode. The e.m.f. Value is determined by the potential difference between the two electrodes. The analyte's concentration and the potential difference is measured by the Nernst equation [129], which is explained as follows:

E_cell=E_0-RT/zF lnQ                      (2)                                                                                                                                

Where Ecell is the e.m.f., Eo is the potential of the standard electrode, R is the gas constant, T is the temperature in Kelvin, z is no of charge of the electrode reaction, F is the Faraday constant, and Q is ion concentration ratio of the anode to cathode.”

We have also added a graphical abstract to highlights all the major sections which was explained in the review paper. It will help the readers to overview of the whole concept of biosensors and their application for water quality monitoring.

Reviewer 2 Report

This paper reports the recent advancements in electrochemical biosensors for monitoring water quality. In general, the manuscript is well-organized, and logically laid out. This paper is possibly publishable. For improving a manuscript, it is advisable to address the following comments:

  1. In lines 106-111 of Page 3, various types of biosensors using different mechanisms were introduced. Recent representative references should be cited for each type. For example, the optical biosensor integrated with antibody and functional polymer could be cited to enrich the introduction section and support your opinion. (https://doi.org/10.1016/j.bios.2019.03.024)
  2. In terms of water quality monitoring, please explain more about which chemical pollutants in the water this paper focuses on and the disadvantages of other techniques in the introduction section.
  3. In Table 2, the response time should be added.
  4. The authors should check carefully across the entire manuscript for the typo, grammar, etc.

Author Response

Reviewer 2

This paper reports the recent advancements in electrochemical biosensors for monitoring water quality. In general, the manuscript is well-organized, and logically laid out. This paper is possibly publishable. For improving a manuscript, it is advisable to address the following comments:

In lines 106-111 of Page 3, various types of biosensors using different mechanisms were introduced. Recent representative references should be cited for each type. For example, the optical biosensor integrated with antibody and functional polymer could be cited to enrich the introduction section and support your opinion. (https://doi.org/10.1016/j.bios.2019.03.024)

Reply: Thanks for your insightful advice. We have added the recent representative references, including electrochemical [Xiao G ,  Song Y ,  Zhang Y , et al. Microelectrode Arrays Modified with Nanocomposites for Monitoring Dopamine and Spike Firings Under Deep Brain Stimulation in Rat Models of Parkinson's Disease[J]. ACS Sensors, 2019, 4(8).], (iii) piezoelectric [Tian Y ,  Zhu P ,  Chen Y , et al. Piezoelectric aptasensor with gold nanoparticle amplification for the label-free detection of okadaic acid[J]. Sensors and Actuators B Chemical, 2021, 346:130446.], (ii) optical [Yang F ,  Chang T L ,  Liu T , et al. Label-free detection of Staphylococcus aureus bacteria using long-period fiber gratings with functional polyelectrolyte coatings[J]. Biosensors & Bioelectronics, 2019, 133:147-153.], and (iv) thermal biosensors [Khorshid M ,  Sichani S B ,  Cornelis P , et al. The hot-wire concept: Towards a one-element thermal biosensor platform[J]. Biosensors & Bioelectronics, 2021(552):113043.].

In terms of water quality monitoring, please explain more about which chemical pollutants in the water this paper focuses on and the disadvantages of other techniques in the introduction section.

Reply: Thanks for your comment. This review mainly focused on the monitoring of heavy metals, nutrients, organic pollutants, biochemical oxygen demand and microorganisms.

The following statements are in the introduction section for clear understanding:

“The characteristics of water pollution are comprised of their physical presence, chemical parameters, and richness of microorganisms. The concentration and composition of ingredients in water differ extensively. They can be categorized into four distinct classifications such as (i) inorganic chemicals, (ii) nutrients, (iii) micro-organisms’ pollution, and (iv) organic pollutants. They can bring about harmful ecological consequences, for example, interference of internal secretion and hor-mone systems, stimulation of genotoxicity and cytotoxicity, and hazardous effects [5]. The strength of ingredients in water is essential for selecting, designing, and opera-tional treatment processes and recycling waste. The variable quantity of contami-nants in effluent over time also increases the attention to emerging technologies for monitoring the water, applying reasonably priced and real-time approaches [6]. This review mainly focused on the monitoring ofmonitoring heavy metals, nutrients, organic pollutants, biochemical oxygen demand and microorganisms. Heavy metals in soil and water are considered environmental contaminants with elevated toxicity, easy accretion, and complicated degradation [7]. Nutrients bring about water eu-trophication. Organic pollutants, particularly persistent organic pollutants (POPs), have harshly harmful impacts on human health and the environment with their complex degradation and potential bioaccumulation [8]. The biochemical oxygen demand (BOD) is the essential supervisory index to measure organic water con-tamination and demonstrate water quality [9,10]. Water quality monitoring is critical and closely related to our life and production.”

Disadvantages of other techniques in the introduction section were illustrated as follows:

 “Conventional analytical techniques or laboratory-based procedures such as gas chromatography (GC), high-performance liquid chromatography (HPLC), atomic absorption spectroscopy (AAS), atomic fluorescence spectrometry (AFS), and inductively coupled plasma mass spectrometry (ICP-MS) and mass spectroscopy (MS) are sensitive, precise, and consistent.  They are regularly used to measure water parameters with the help of trained operators. However, they are involved with bulky and costly instrumentation, take much time for sample preparation, and are unsuitable for in situ measurements, especially requiring trained operators' help and transporting the water samples to the laboratory for assessment [11-13]. Addi-tionally, they can notcannot asses the accumulative toxicity or nutrient value of multiple chemicals or pollutants in a sample, which is a crucial objective of water quality monitoring applications [14]. Many property indicators are regularly used to determine the different qualities of water for settling or recycling. Many of them are laboratory-based techniques, which require frequent sample data sampling for pre-treatment, and consequently, the methods are sluggish and expensive [1,15]. These characteristics encourage developing new technologies that are more low-cost, portable, sensitive, and efficient in the on-site real-time detection of mul-ti-contaminants containing a wide variety of materials [16,17].”

We have added the following statements to show the disadvantages of other method such as optical method as follows:

“In comparison with optical methods, electrochemical transduction has advantages for analyzing turbid samples because it is non-sensitive to light. For optical sensing, they are likely to be interference from environmental effects, costly and susceptible to physical damage.”

In Table 2, the response time should be added.The authors should check carefully across the entire manuscript for the typo, grammar, etc.

Reply: Thank you for your comment. The response time is added in Table 2. And we have went through the manuscript carefully to correct all the typos and grammar.

Reviewer 3 Report

Current manuscript entitled “Recent advancements in electrochemical biosensors for monitoring the water quality” by Hui “et al” demonstrated on the advances made in electrochemical biosensors for water quality monitoring. The presentation is good, manuscript written well and can be accepted after addressing the following comments.

1.      Provide advantages of electrochemical sensors.

2.      Improve the image quality of Figure 1.

3.      In the abstract please provide the innovation of the current work.

4.      In the manuscript, authors mentioned on the portable devices. However, in the introduction section potential information on the portable devices and on-site sensing strategies is missing. Some potential literature is dedicated on the on-site detection of pollutants and hazardous constituents that can provide a basic idea. The authors should discuss these in the manuscript.

https://doi.org/10.1016/j.ccr.2021.214305

Biosensors 2022, 12(5), 259; https://doi.org/10.3390/bios12050259

Author Response

Reviewer 3

Current manuscript entitled “Recent advancements in electrochemical biosensors for monitoring the water quality” by Hui “et al” demonstrated on the advances made in electrochemical biosensors for water quality monitoring. The presentation is good, manuscript written well and can be accepted after addressing the following comments.

Reply: Thanks for your encouraging comments.

  1. Provide advantages of electrochemical sensors.

Reply: Thanks for your comment. The advantages are explained in the introduction section as the following manner. We have also included the disadvantages of optical based sensor over electrochemical biosensors.

“An electrochemical biosensor is based on the interaction among the immobilized bio-recognition element on its surface with binding molecules (the analyte of interest) and generating the changes in electrochemical properties, further translating into a meaningful electrical signal. The electrochemical methods offer rapid detection, fabrication, excellent sensitivity, and low cost. In comparison with optical methods, electrochemical transduction has advantages for analyzing turbid samples because it is non-sensitive to light. Moreover, by operating at a wide range of potential, it is possible to simultaneously determine multiple analytes with different electrochemical potentials. Electrochemical biosensors' efficiency in monitoring water pollutants' presence relied on bio-recognition elements, transducers, and immobilization techniques, which offer us the classification criterion. In comparison with optical methods, electrochemical transduction has advantages for analyzing turbid samples because it is non-sensitive to light. For optical sensing, they are likely to be interference from environmental effects, costly and susceptible to physical damage.”

  1. Improve the image quality of Figure 1.

Reply: Thanks for your comment. The figure 1 has some copyright issues and therefore the similar images are replaced, and overall quality is improved as follows.

  1. In the abstract please provide the innovation of the current work.

Reply: Thanks for your comment. We have included the following statements in the abstract for highlighting the objective of this review paper.

“This review paper is on the basic concepts of electrochemical biosensors, and their applications in various water quality monitoring, such as, inorganic chemicals, nutrients, microorganisms’ pollution, and organic pollutants especially for developing real-time/online detection systems. The basic concepts of electrochemical biosensors, different surface modification techniques, bio-recognition elements (BRE), detection methods, and specific real-time water quality monitoring applications are reviewed thoroughly in this article.”

  1. In the manuscript, authors mentioned on the portable devices. However, in the introduction section potential information on the portable devices and on-site sensing strategies is missing. Some potential literature is dedicated on the on-site detection of pollutants and hazardous constituents that can provide a basic idea. The authors should discuss these in the manuscript.

https://doi.org/10.1016/j.ccr.2021.214305

Biosensors 2022, 12(5), 259; https://doi.org/10.3390/bios12050259

Reply: Thanks for your suggestions and the recent papers. We have discussed in the introduction sections regarding the basic ideas of on-site detection of pollutants for other applications and their problems. They are as follows:

“Conventional analytical techniques or laboratory-based procedures such as gas chromatography (GC), high-performance liquid chromatography (HPLC), atomic absorption spectroscopy (AAS), atomic fluorescence spectrometry (AFS), and inductively coupled plasma mass spectrometry (ICP-MS)  are sensitive, precise, and consistent. They are regularly used to measure water parameters with the help of trained operators. However, they are involved with bulky and costly instrumenta-tion, take much time for sample preparation, and are unsuitable for in situ meas-urements, especially requiring trained operators' help and transporting the water samples to the laboratory for assessment [11-13]. Additionally, they cannot asses the accumulative toxicity or nutrient value of multiple chemicals or pollutants in a sample, which is a crucial objective of water quality monitoring applications [14]. Many property indicators are regularly used to determine the different qualities of water for settling or recycling. Many of them are laboratory-based techniques, which require frequent sample data sampling for pretreatment, and consequently, the methods are sluggish and expensive [1,15]. These characteristics encourage devel-oping new technologies that are more low-cost, portable, sensitive, and efficient in the on-site real-time detection of multi-contaminants containing a wide variety of materials [16,17]. The major challenges of developing portable biosensing device are inadequate sensitivity and poor selectivity during the on-site detection. The great level of noises can come on chemical components level from the sampling field and ambient environments can be variable due to the harsh environments or diurnal variations. These are the major obstacles where the researchers are putting lot of attentions on how to avoid these for generating a reliable and portable biosensing output signal. The portable biosensing method is successfully utilizing for other applications, such as  pesticide residues in fruits and vegetables [18], POC Detec-tion for biomedical application [19], chemical and biological pollutants in water [20].”
